# Ultra-Widefield Retinal Imaging for Analyzing the Association Between Types of Pathological Myopia and Posterior Staphyloma

**DOI:** 10.3390/jcm8101505

**Published:** 2019-09-20

**Authors:** Ririko Mimura, Kiwako Mori, Hidemasa Torii, Norihiro Nagai, Misa Suzuki, Sakiko Minami, Yoko Ozawa, Toshihide Kurihara, Kazuo Tsubota

**Affiliations:** 1Department of Ophthalmology, Ichikawa General Hospital, Tokyo Dental College, 5-11-13, Sugano, Ichikawa, Chiba 272–8513, Japan; 2Department of Ophthalmology, Keio University School of Medicine, 35 Shinanomachi, Shinjuku-ku, Tokyo 160-8582, Japan; 3Laboratory of Photobiology, Keio University School of Medicine, 35 Shinanomachi, Shinjuku-ku, Tokyo 160-8582, Japan; 4Department of Ophthalmology, Yokohama City University School of Medicine, 3-9 Fukuura, Kanazawa-ku, Yokohama, Kanagawa 236-0004, Japan

**Keywords:** high myopia, staphyloma, pathological myopia, axial length

## Abstract

High myopia may develop to pathologic myopia, which brings severe visual impairment; however, the etiology is not fully understood. We, therefore, investigated the relationship between the presence of posterior staphyloma and posterior ocular disorders by assessing the patients with high myopia. A retrospective study was performed for the patients, who have more than 26 mm of the axial length and of whom fundus photography was taken with an ultra-widefield retinal imaging system. The objectives were 138 cases encompassing 229 eyes. In 138 cases, 91 were bilateral and 47 were unilateral. The averages ± SD of axial length of bilateral and unilateral were 28.8 ± 2.2 mm, 27.3 ± 1.2 mm, respectively, showing statistically significant difference. The number of eyes with and without posterior staphyloma were 107 (46.7%) and 122 (53.3%), respectively. Retinal detachment and retinal breaks are more observed in cases without posterior staphyloma (*p* = 0.017). Myopic choroidal neovascularization (mCNV) (*p* = 0.002), chorioretinal atrophy (*p* < 0.001), retinoschisis (*p* < 0.001), and optic neuropathy (*p* = 0.038) are more often seen in cases with posterior staphyloma. In conclusion, the prevalence rates of myopic choroidal neovascularization, retinal choroidal atrophy, and optic neuropathy were significantly higher with posterior staphyloma. The rate of periocular disorders such as retinal detachment was significantly higher without posterior staphyloma. These results indicate associations between types of pathological myopia and presence or absence of posterior staphyloma analyzed by ultra-widefield retinal imaging.

## 1. Introduction

The prevalence of myopia has been recently increasing, and it is remarkable especially in the East Asian countries including Japan. With the increase in the number of myopia cases, it is anticipated that the prevalence of high myopia will increase as well. As many as 0.9 billion people, approximately 10% of the world population, are estimated to have high myopia [1]. Along with the increase of high myopia population, ocular disorders associated with high myopia are also expected to increase. Therefore, there are some concerns regarding medical and surgical costs, which stimulate the understanding of the mechanism of high myopia and its prevention. High myopia gives rise to several myopia-related ocular disorders leading to severe visual impairments by elongation of the axial length which is the anteroposterior length of the eye [2]. High myopia is often bilateral, irreversible, and common in people in their prime; thus, it is critical condition from the socioeconomical standpoint [3,4,5]. Although studies regarding myopia, high myopia, and pathologic myopia are enthusiastically pursued, there are practically few measures to prevent the progression of myopia.

Posterior staphyloma, which is the posterior outpouching of the wall of the eye, is an important component of the diagnosis of pathologic myopia; indeed, it is one of the hallmarks of pathologic myopia [2,6,7]. Recently, it has successfully been demonstrated that 3-dimensional analysis of the shape of the eye using 3D-MRI provided the shape of the fundus and whole eyeball [8]. Due to the development of imaging studies, the classification of posterior staphyloma initially determined by the fundoscopy was revised and replaced by the classification with wide-angle fundus photographs and 3D-MRI [2]. Because of these changes, the association between abnormal shapes of the eye in high myopia and the progression of visual impairment has been much more established.

The aim of this study is to investigate associations between types of pathological myopia and presence or locations of staphyloma analyzed by the ultra-widefield imaging technique.

## 2. Materials and Methods

### 2.1. Ethical Guidelines

This study was a hospital-based retrospective study and was conducted in compliance with the Declaration of Helsinki, Ethical Guidelines for Medical and Health Research Involving Human Subjects, and local regulatory requirements and also under the approval of all study institutional review boards (IRB) and ethics committees. This trial was approved by the IRB of Keio University School of Medicine (approval no. 20180189).

### 2.2. Subjects

Data were collected and analyzed from the medical records of the patients who visited Department of Ophthalmology at Keio University Hospital for the duration 1 July 2017 to 30 June 2018. The ocular physical examination was performed as usual in the conventional way, and unnecessary imaging studies or laboratory exams were not performed for this study. Although refraction and axial length are indicators of myopia, refraction would be changed from the naïve status in some cases such as histories of cataract surgery. Therefore, we used axial length but not refraction as an indicator of myopia in this study. The subjects had 26 mm or longer of the axial length measured by optical biometry (IOLMaster 500, Zeiss, Jena, Germany) and on whom ultra-widefield retinal imaging system (Optos California, Nikon, Tokyo, Japan) was performed.

### 2.3. Data Analysis

The data related to patients’ information and medical history were retrieved from medical records. The classification of posterior staphyloma was performed with Curtin’s and Ohno–Matsui’s classifications using the ultra-widefield retinal imaging system. The measurement was carried out by two independent ophthalmologists in a blind fashion.

### 2.4. Statistics

Data are presented as mean ± standard deviation. All data obtained were used for statistical analysis. Independent *t*-tests were used for the comparison of age and axial length for the parametric data. The chi-square test was used for categorical variables. Statistical analyses were conducted using the SPSS version 23.0 for Windows (IBM, Armonk, NY, USA), and statistical significance was defined as *p* < 0.05.

## 3. Results

### 3.1. Characteristics of The Patients

Among patients who visited Department of Ophthalmology at Keio University Hospital for the specific, 1025 cases examined their axial lengths utilizing the optical biometry. Out of 1025 patients, 332 were shown to have 26 mm or more of the axial length, of which 138 patients had a photo of the ultra-widefield retinal imaging system (Figure 1).

The average of age and axial length was 63.1 ± 13.5 years old and 28.1 ± 2.2 mm, respectively (Table 1). Of 229 eyes, 107 (46.7%) eyes were shown to have posterior staphyloma, and there were 51 male eyes and 56 female eyes, showing statistically female dominance (*p* = 0.033).

### 3.2. Comparison of Bilateral and Unilateral Axial Elongation

The objectives were 138 cases encompassing 229 eyes. Out of 138 cases, 91 cases were bilateral and 47 were unilateral (27 right eyes and 20 left eyes). The average of age was 63.0 ± 14.0 years old in bilateral and 63.4 ± 12.4 years old in unilateral, and there was no significant difference between the groups (*p* = 0.881). Regarding the gender difference, there was no statistical significance (male: 53.80%, female: 46.2% in bilateral; male: 61.7%, female: 38.3% in unilateral, *p* = 0.469). Axial lengths of bilateral and unilateral were 28.8 ± 2.2 mm and 27.3 ± 1.2 mm, respectively, showing a statistically significant difference (*p* < 0.001) (Table 2).

### 3.3. Relationship of Axial Length to Factors

The measured axial lengths were divided into quartiles. The relationship of axial length to gender, presence of posterior staphyloma, classification of posterior staphyloma, and past medical history was analyzed (Table 3). None was shown to be related to the axial length with statistical significance.

### 3.4. Comparison in the Presence of Posterior Staphyloma

The numbers of eyes with and without posterior staphyloma were 107 (46.7%) and 122 (53.3%), respectively. As for gender difference in cases with posterior staphyloma, female (52.3%) was dominant compared to male (47.7%) (*p* = 0.033) (Table 1). The average of age was 67.8 ± 12.6 years old in cases with posterior staphyloma, which is significantly higher than 59.4 ± 13.1 years old in cases without posterior staphyloma (*p* < 0.001). The average of axial length was 28.4 ± 2.1 mm with posterior staphyloma and 28.5 ± 2.1 mm without posterior staphyloma, showing no statistical difference (*p* = 0.783).

Regarding relationship of past medical history, peripheral retinal disorders such as retinal detachment and retinal breaks are more observed in cases without posterior staphyloma (*p* = 0.017). Posterior pole lesions including myopic choroidal neovascularization (mCNV), chorioretinal atrophy, retinoschisis, and optic neuropathy are more often seen in cases with posterior staphyloma (Table 4, Figure 2).

### 3.5. Classification of Posterior Staphyloma and Disorders

Among cases with posterior staphyloma, the relationship between the classifications of posterior staphyloma and disorders was investigated. In both Curtin’s and Ohno–Matsui’s classifications, there was no relationship between the type of classification and disorders (Table 5, Figure 3).

## 4. Discussion

This study demonstrated that in the eyes with 26 mm or more of the axial length bilateral cases were more prevalent and the axial length of bilateral cases was longer (Table 2), and that lesions in the posterior pole are more prevalent with posterior staphyloma and the periocular disorders are more prevalent without posterior staphyloma (Table 4). There was no relationship between the axial length and ocular disorders in the current cases (Table 3). The classification of posterior staphyloma was not associated with ocular disorders (Table 5).

It was previously reported that posterior staphyloma was related to low visual acuity and high rates of macular hole retinal detachment (MHRD), mCNV, and chorioretinal atrophy. Among them, MHRD is more common especially in posterior staphyloma type II [2,9]. The current study showed similar findings with the previous reports such as associations between posterior staphyloma and posterior pole disorders. Exceptionally, no significant association was observed between classifications of posterior staphyloma and types of ocular disorders.

It has been speculated that excessive elongation of the axial length could lead to mechanical damages in the retina and the optic nerve by stretching the wall of the eyeball [10]. As demonstrated in this study, retinoschisis, mCNV, chorioretinal atrophy, and optic neuropathy are more often observed in cases with posterior staphyloma. The current data supported the hypothesis that the posterior pole disorders may be developed as a result of damages in the retina and the optic nerve by stretching the part of the wall of the eyeball caused by posterior staphyloma.

There is a report showing cases of nonperfusion area in the peripheral retina with ocular elongation in the anterior part of the equator in cases with high myopia [11]. Myopic axial elongation is a continuous process of emmetropization, and a continuous elongation of the peripheral region might occur in highly myopic eyes when the axial length increases [11]. In this study, more periocular disorders such as retinal detachment and retinal breaks are seen in the cases without posterior staphyloma supporting that retinal breaks may be caused by elongation of the region between the most peripheral and the equator.

This study has some limitations. First, there was an inclusion bias because of the hospital-based study. Thus, the prevalence rate of diseases may have been higher than the actual rate because most of the patients who need to undergo fundus photography are expected to have some retinal disease. Second, the subjects less than 26 mm of the axial length were excluded. There are cases who have posterior staphyloma with less than 26 mm of axial lengths in our record.

There are two types of axial elongation, the one type with a change from the equator to peripheral, and the other type with a change in the posterior pole. Peripheral retinal diseases in the cases without posterior staphyloma often happen and are thought to be through thinning of the peri-retinal region. Posterior pole retinal diseases and optic neuropathy in the cases with posterior staphyloma often happen and are through thinning of the choroid, circulatory disturbance, or traction of the macula.

Recent advancement of wide or ultra-wide field imaging systems enabled us to evaluate most areas of the retina providing new perspectives on diagnosis and treatment of retinal diseases [12,13]. This study demonstrated the superiority of this imaging technique to show the presence and location of posterior staphyloma. Furthermore, the relationship between finding posterior staphyloma and ocular disorders which are seen in pathological myopia raised the possibility of application of this imaging to clinical settings. It is crucial to detect such abnormalities as early as possible in high myopia eyes since advances in myopic control have been remarkable. The results of the study indicate that when posterior staphyloma is not observed in high myopia patients with ultra-widefield imaging, further careful investigation of the fundus with pupil dilation can be recommended to investigate peripheral degeneration and retinal breaks. Especially for the patients with high myopia, this imaging could be helpful to provide possibilities of finding disorders related to pathological myopia.

## Figures and Tables

**Figure 1 jcm-08-01505-f001:**
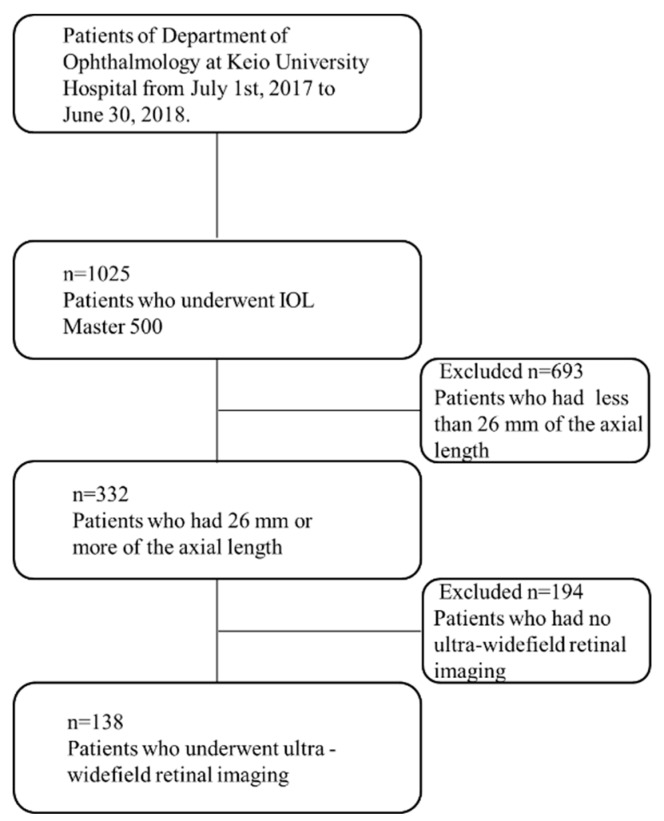
Flowchart of selecting the patients in this study.

**Figure 2 jcm-08-01505-f002:**
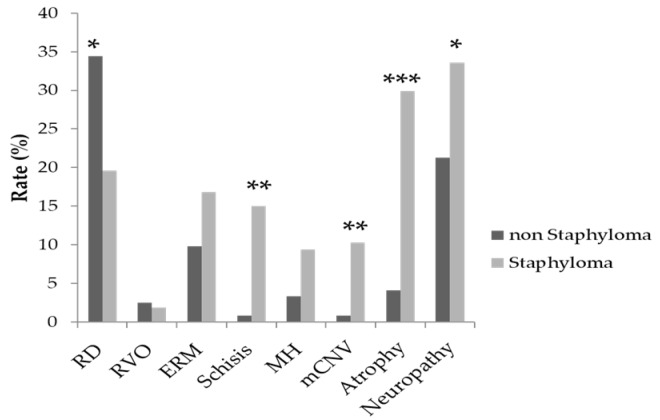
The rate of presence of posterior staphyloma in each ocular disorder. RD has a lower presence rate of posterior staphyloma, whereas schisis, mCNV, atrophy, and neuropathy have a higher presence rate of posterior staphyloma. RD: retinal detachment and retinal tear; RVO: central retinal vein occlusion and branch retinal vein occlusion; ERM: epiretinal membrane; Schisis: retinoschisis; MH: macular hole; mCNV: myopic choroidal neovascularization; Atrophy: chorioretinal atrophy; Neuropathy: glaucoma and myopic optic neuropathy. (* *p* < 0.05, ** *p* < 0.01, *** *p* < 0.001).

**Figure 3 jcm-08-01505-f003:**
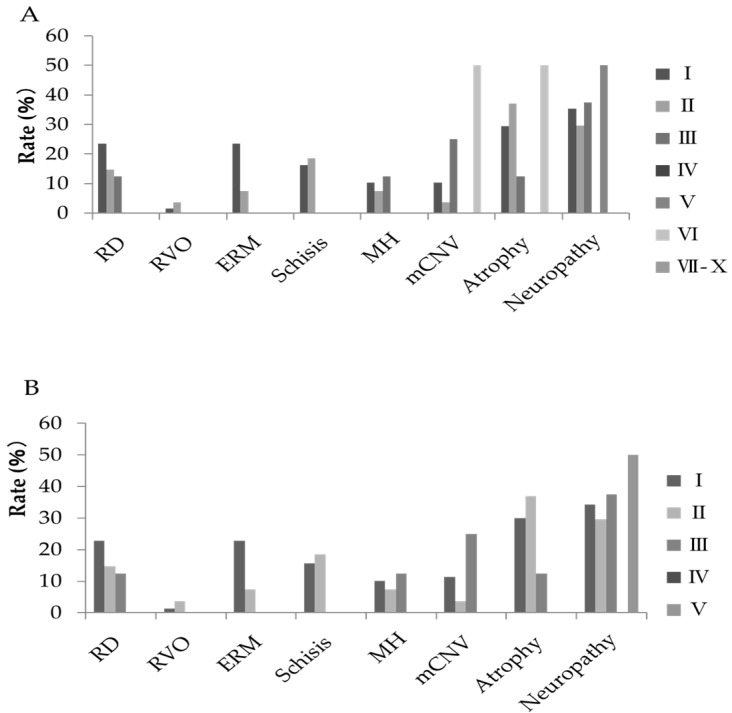
Classification of posterior staphyloma in each ocular disorder in two classifications, Curtin’s classification (**A**) and Ohno–Matsui’s classification (**B**). Among the classes of posterior staphyloma, no association is observed in each disorder in both classifications. Staphyloma: posterior staphyloma; RD: retinal detachment and retinal tear; RVO: central retinal vein occlusion and branch retinal vein occlusion; ERM: epiretinal membrane; Schisis: retinoschisis; MH: macular hole; mCNV: myopic choroidal neovascularization; Atrophy: chorioretinal atrophy; Neuropathy: glaucoma and myopic optic neuropathy.

**Table 1 jcm-08-01505-t001:** Characteristics of high myopia patients.

	All	Men	Women	*p* Value	
Subjects *n* (%)	138	78 (56.4)	60 (43.6)		
Age (yrs)	63.1 ± 13.5	62.1 ± 12.5	64.5 ± 14.7	0.306	†
AXL (mm)	28.1 ± 2.2	28.1 ± 2.3	28.2 ± 2.1	0.822	†
Staphyloma *n* (%)	107 (46.7)	51 (47.7)	56 (52.3)	0.033	††

Data represent means ± standard deviations, AXL: axial length, Staphyloma: posterior staphyloma, †: student *t* test, ††: chi-square test.

**Table 2 jcm-08-01505-t002:** Comparison of bilateral and unilateral axial elongation.

	Bilateral	Unilateral	*p* Value	
All *n* (%)	91 (65.9)	47 (34.1)		
Men *n* (%)	49 (53.8)	29 (61.7)	0.469	†
Women *n* (%)	42 (46.2)	18 (38.3)		
Age (yrs)	63.0 ± 14.0	63.4 ± 12.4	0.881	††
AXL (mm)	28.8 ± 2.2	27.3 ± 1.2	<0.001	††

Data represent means ± standard deviations, AXL: axial length, Bilateral: both eyes with AXL not less than 26 mm, Unilateral: one eye with AXL not less than 26 mm, †: chi-square test, ††: student *t* test.

**Table 3 jcm-08-01505-t003:** Relationship of axial length to factors.

		AXL				*p* Value
		26.00 to 26.79 mm	26.80 to 27.67 mm	27.68 to 29.76 mm	29.77 mm≤	
Subjects	All	57	58	57	57	
	Men *n* (%)	33 (57.9)	34 (58.6)	24 (42.1)	36 (63.2)	0.119
	Women *n* (%)	24 (42.1)	24 (41.4)	33 (57.9)	21 (36.8)	
Staphyloma	All *n* (%)	27 (47.4)	30 (51.7)	23 (40.4)	27 (47.4)	0.675
	Men *n* (%)	16 (48.5)	12 (35.3)	6 (25.0)	17 (47.2)	0.228
	Women *n* (%)	11 (45.8)	18 (75.0)	17 (51.5)	10 (47.6)	0.150
Curtin (*n*)	Ⅰ	14	17	14	23	
	Ⅱ	9	8	7	3	
	Ⅲ	3	4	0	1	
	Ⅳ	0	0	0	0	
	Ⅴ	1	0	1	0	
	Ⅵ	0	1	1	0	
	Ⅶ-Ⅹ	0	0	0	0	0.391
Ohno–Matsui (*n*)	Ⅰ	14	18	15	23	
	Ⅱ	9	8	7	3	
	Ⅲ	3	4	0	1	
	Ⅳ	0	0	0	0	
	Ⅴ	1	0	1	0	0.334
Disorders	RD *n* (%)	17 (27.8)	17 (29.3)	16 (28.1)	13 (22.8)	0.784
	RVO *n* (%)	0 (0.0)	1 (1.7)	0 (0.0)	4 (7.0)	0.032
	ERM *n* (%)	3 (5.3)	5 (8.6)	8 (14.0)	14 (24.6)	0.013
	Schisis *n* (%)	1 (1.8)	3 (5.2)	5 (8.8)	8 (14.0)	0.076
	MH *n* (%)	4 (7.0)	4 (6.9)	1 (1.8)	5 (8.9)	0.425
	mCNV *n* (%)	1 (1.8)	6 (10.3)	4 (7.0)	1 (1.8)	0.102
	Atrophy *n* (%)	7 (18.9)	13 (35.1)	10 (27.0)	7 (18.9)	0.388
	Neuropathy *n* (%)	22 (38.6)	15 (25.9)	11 (19.3)	14 (24.6)	0.122

AXL: axial length; Staphyloma: posterior staphyloma; Curtin: Curtin’s classification; Ohno–Matsui: Ohno–Matsui’s classification; RD: retinal detachment and retinal tear; RVO: central retinal vein occlusion and branch retinal vein occlusion; ERM: epiretinal membrane; Schisis: retinoschisis; MH: macular hole; mCNV: myopic choroidal neovascularization; Atrophy: chorioretinal atrophy; Neuropathy: glaucoma and myopic optic neuropathy.

**Table 4 jcm-08-01505-t004:** Comparison in the presence of posterior staphyloma.

	Non Staphyloma	Staphyloma	*p* Value	
Age (yrs)	59.4 ± 13.1	67.8 ± 12.6	<0.001	†
AXL (mm)	28.5 ± 2.1	28.4 ± 2.1	0.783	†
Disorders				
RD (*n*)	42	21	0.017	††
RVO (*n*)	3	2	1.000	††
ERM (*n*)	12	18	0.169	††
Schisis (*n*)	1	16	<0.001	††
MH (*n*)	4	10	0.094	††
mCNV (n)	1	11	0.002	††
Atrophy (*n*)	5	32	<0.001	††
Neuropathy (*n*)	26	36	0.038	††

Data represent means ± SDs, †: student *t* test, ††: chi-square test; AXL: axial length; Staphyloma: posterior staphyloma; RD: retinal detachment and retinal tear; RVO: central retinal vein occlusion and branch retinal vein occlusion; ERM: epiretinal membrane; Schisis: retinoschisis; MH: macular hole; mCNV: myopic choroidal neovascularization; Atrophy: chorioretinal atrophy; Neuropathy: glaucoma and myopic optic neuropathy.

**Table 5 jcm-08-01505-t005:** Classification of posterior staphyloma and disorders. (A) Curtin’s classification. (B) Ohno–Matsui’s classification.

(**A**)									
	**Staphyloma (*n*)**	**Curtin’s Classification**							***p* Value**
		**I**	**II**	**III**	**IV**	**V**	**VI**	**VI-X**	
total (*n*)	107	68	27	8	0	2	2	0	
RD *n* (%)	21	16 (23.5)	4 (14.8)	1 (12.5)	0 (0.0)	0 (0.0)	0 (0.0)	0 (0.0)	0.683
RVO *n* (%)	2	1 (1.5)	1 (3.7)	0 (0.0)	0 (0.0)	0 (0.0)	0 (0.0)	0 (0.0)	0.941
ERM *n* (%)	18	16 (23.5)	2 (7.4)	0 (0.0)	0 (0.0)	0 (0.0)	0 (0.0)	0 (0.0)	0.176
Schisis *n* (%)	16	11 (16.2)	5 (18.5)	0 (0.0)	0 (0.0)	0 (0.0)	0 (0.0)	0 (0.0)	0.652
MH *n* (%)	10	7 (10.4)	2 (7.4)	1 (12.5)	0 (0.0)	0 (0.0)	0 (0.0)	0 (0.0)	0.949
mCNV *n* (%)	11	7 (10.3)	1 (3.7)	2 (25.0)	0 (0.0)	0 (0.0)	1 (50.0)	0 (0.0)	0.147
Atrophy *n* (%)	32	20 (29.4)	10 (37.0)	1 (12.5)	0 (0.0)	0 (0.0)	1 (50.0)	0 (0.0)	0.548
Neuropathy *n* (%)	36	24 (35.3)	8 (29.6)	3 (37.5)	0 (0.0)	1 (50.0)	0 (0.0)	0 (0.0)	0.812
(**B**)									
	**Staphyloma (*n*)**	**Ohno–Matsui’s Classification**							***p* Value**
		**I**	**II**	**III**	**IV**	**V**			
total (*n*)	107	70	27	8	0	2			
RD *n* (%)	21	16 (22.9)	4 (14.8)	1 (12.5)	0 (0.0)	0 (0.0)			0.658
RVO *n* (%)	2	1 (1.4)	1 (3.7)	0 (0.0)	0 (0.0)	0 (0.0)			0.859
ERM *n* (%)	18	16 (22.9)	2 (7.4)	0 (0.0)	0 (0.0)	0 (0.0)			0.135
Schisis *n* (%)	16	11 (15.7)	5 (18.5)	0 (0.0)	0 (0.0)	0 (0.0)			0.560
MH *n* (%)	10	7 (10.1)	2 (7.4)	1 (12.5)	0 (0.0)	0 (0.0)			0.926
mCNV *n* (%)	11	8 (11.4)	1 (3.7)	2 (25.0)	0 (0.0)	0 (0.0)			0.324
Atrophy *n* (%)	32	21 (30.0)	10 (37.0)	1 (12.5)	0 (0.0)	0 (0.0)			0.446
Neuropathy *n* (%)	36	24 (34.3)	8 (29.6)	3 (37.5)	0 (0.0)	1 (50.0)			0.919

Staphyloma: posterior staphyloma; RD: retinal detachment and retinal tear; RVO: central retinal vein occlusion and branch retinal vein occlusion; ERM: epiretinal membrane; Schisis: retinoschisis; MH: macular hole; mCNV: myopic choroidal neovascularization; Atrophy: chorioretinal atrophy; Neuropathy: glaucoma and myopic optic neuropathy; Staphyloma: posterior staphyloma; RD: retinal detachment and retinal tear; RVO: central retinal vein occlusion and branch retinal vein occlusion; ERM: epiretinal membrane; Schisis: retinoschisis; MH: macular hole; mCNV: myopic choroidal neovascularization; Atrophy: chorioretinal atrophy; Neuropathy: glaucoma and myopic optic neuropathy.

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
