# Peer review of "Ultra-Widefield Retinal Imaging for Analyzing the Association Between Types of Pathological Myopia and Posterior Staphyloma"

_jcm, 2019, doi:10.3390/jcm8101505_

Round 1
Reviewer 1 Report
The manuscript reported the use of ultra-widefield retinal imaging to detect high-myopia related complications. They tried to correlate staphyloma with posterior ocular disorders. They found mCNV, chorioretinal atrophy, retinoschisis and optic neuropathy are associated with posterior staphyloma.
It has been noted that staphyloma is associated with mCNV, chorioretinal atrophy, retinoschisis and optic neuropathy. The reviewer finds no new information provided by this manuscript.
They used a relatively new technique which is ultra-wide retinal imaging system to detect the pathological complications of high-myopia. By using this technique (machine), they concluded that "ultra-widefield retinal imaging to predict complication of high-myopia", which is not true. The imaging technique itself can't PREDICT complications but the staphyloma in the high-myopic eyes.
Author Response
Point-by-point responses to the comment from Reviewer #1
Point #1
The manuscript reported the use of ultra-widefield retinal imaging to detect high-myopia related complications. They tried to correlate staphyloma with posterior ocular disorders. They found mCNV, chorioretinal atrophy, retinoschisis and optic neuropathy are associated with posterior staphyloma.
It has been noted that staphyloma is associated with mCNV, chorioretinal atrophy, retinoschisis and optic neuropathy. The reviewer finds no new information provided by this manuscript.
Response: We greatly appreciate your precious comments regarding our manuscript. The point of this study is comparison of the findings of posterior staphyloma by using an ultra-widefield retinal imaging which is capable of clearly visualizing the entire fundus with types of ocular disorders seen in pathological myopia. This imaging technique enables to photographically evaluate the presence and the location of posterior staphyloma. Therefore, the study demonstrated that merely the presence of posterior staphyloma gave the possibility of eye disorders, whereas the location of posterior staphyloma was irrelevant to them. According to these results, when posterior staphyloma is not observed in high myopia patients with the ultra-widefield imaging, further careful investigation of the fundus with pupil dilation can be recommended to investigate peripheral degeneration and retinal breaks. Since patients with high myopia and no remarkable symptoms do not consult ophthalmologists so often, the expansion of application of this imaging technique could be of much help to provide possibilities of finding disorders related to pathological myopia. Considering these values of this study, we revised the description in the discussion section as follows.
Line 218:
Deleted; Our study suggested that ultra-widefield imaging could be applied to the evaluation and the prognosis of high myopia.
Revised; It is crucial to detect such abnormalities as early as possible in high myopia eyes because advances in myopic control have been remarkable. The results of the study indicate that when posterior staphyloma is not observed in high myopia patients with the ultra-widefield imaging, further careful investigation of the fundus with pupil dilation can be recommended to investigate peripheral degeneration and retinal breaks. Especially for the patients with high myopia, this imaging could be helpful to provide possibilities of finding disorders related to pathological myopia.
Point #2
They used a relatively new technique which is ultra-wide retinal imaging system to detect the pathological complications of high-myopia. By using this technique (machine), they concluded that "ultra-widefield retinal imaging to predict complication of high-myopia", which is not true. The imaging technique itself can't PREDICT complications but the staphyloma in the high-myopic eyes.
Response: As suggested in your comment, we recognized the manuscript contained fatal misunderstanding stating that the new imaging technique can be used to predict complication of high myopia. We reviewed our entire manuscript and revised/removed the sentences which may make readers think of the possibility of this imaging to predict high myopia complications. We fully revised the sentences including the title concerning the prediction of high myopia.
The followings are revisions we made in accordance with the comments from the reviewer #1.
Likewise, the title "Ultra-widefield retinal imaging to predict complications of high myopia" was inappropriate. Therefore, we would like to be allowed to change the title to "Ultra-widefield retinal imaging for analyzing the association between types of pathological myopia and posterior staphyloma"
Line 2:
Deleted; Ultra-widefield retinal imaging to predict complications of high myopia
Revised; Ultra-widefield retinal imaging for analyzing the association between types of pathological myopia and posterior staphyloma
Line 33:
Deleted: These results suggest that ultra-widefield retinal imaging may be effective to predict complications of high myopia.
Revised: In conclusion, the prevalence rates of myopic choroidal neovascularization, retinal choroidal atrophy, and optic neuropathy were significantly higher with posterior staphyloma. The rate of periocular disorders such as retinal detachment was significantly higher without posterior staphyloma. These results indicate associations between types of pathological myopia and presence or absence of posterior staphyloma analyzed by ultra-widefield retinal imaging.
Line 49:
Deleted: Furthermore, some of complications in pathologic myopia have no therapeutic solutions to date.
Line 224:
Deleted; Our study suggested that ultra-widefield imaging could be applied to the evaluation and the prognosis of high myopia.

Reviewer 2 Report
- L61, P2: The study ‘was conducted to investigate the relationship between the eye structure and pathological myopia’
The aim of the study should be more specific. What the authors mean by ‘eye structure’ and how the ‘relationship’ they investigated could be useful in real-life applications?
- All results were presented as tables, authors should consider using figures and represent 50% of their results graphically.
- It was really difficult to identify the novelty in this study. What is new?
- The study looks like pure statistical analysis without much in-depth discussion about how the findings could be useful.
Author Response
Point-by-point responses to the comment from Reviewer #2
Point #1
- L61, P2: The study ‘was conducted to investigate the relationship between the eye structure and pathological myopia’
The aim of the study should be more specific. What the authors mean by ‘eye structure’ and how the ‘relationship’ they investigated could be useful in real-life applications?
Response: We appreciate your precious comments regarding our manuscript. In this study, we tried to identify the relationship between posterior staphyloma detected by the ultra-widefield imaging and types of pathological myopia. Although this is a cross sectional study to analyze the relationship, the result of the analysis could be helpful to warn the risk especially for the patients with high myopia many of whom do not often consult ophthalmologists. As suggested in your comment, we changed the last part of the introduction to specify the aim of this study.
Line 66:
Deleted; In this study, to seek the mechanism of progression of disorders associated with high myopia by analyzing the structural changes of the eye, a retrospective study using the medical records of the patients who consulted in our institute was conducted to investigate the relationship between the eye structure and pathological myopia.
Revised; The aim of this study is to investigate associations between types of pathological myopia and presence or locations of staphyloma analyzed by the ultra-widefield imaging technique.
Point #2
- All results were presented as tables, authors should consider using figures and represent 50% of their results graphically.
Response: We greatly appreciate your suggestion. We presented figures in addition to tables to make the result visually recognizable.
Added; Figure 2
Figure 2. The rate of presence of posterior staphyloma in each ocular disorder. RD has a lower presence rate of posterior staphyloma, whereas schisis, mCNV, atrophy, and neuropathy has a higher presence rate of posterior staphyloma. RD: retinal detachment and retinal tear; RVO: central retinal vein occlusion and branch retinal vein occlusion; ERM: epiretinal membrane; Schisis: retinoschisis; MH: macular hole; mCNV: myopic choroidal neovascularization; Atrophy: chorioretinal atrophy; Neuropathy: glaucoma and myopic optic neuropathy. (*p < 0.05, **p < 0.01, ***p < 0.001)
Revised; line 150: (Table 4, Figure 2)
Added; Figure 3
Figure 3. Classification of posterior staphyloma in each ocular disorder in two classifications, Curtin's classification (A) and Ohno-Matsui's classification (B). Among the classes of posterior staphyloma, no association is observed in each disorder in both classifications. Staphyloma: posterior staphyloma; RD: retinal detachment and retinal tear; RVO: central retinal vein occlusion and branch retinal vein occlusion; ERM: epiretinal membrane; Schisis: retinoschisis; MH: macular hole; mCNV: myopic choroidal neovascularization; Atrophy: chorioretinal atrophy; Neuropathy: glaucoma and myopic optic neuropathy.
Revised; line 165: (Table 5, Figure 3)
Point #3
- It was really difficult to identify the novelty in this study. What is new?
Response: The point of this study is comparison between the findings of posterior staphyloma by using an ultra-widefield retinal imaging which is capable of clearly visualizing the entire fundus and types of ocular disorders seen in pathological myopia. This imaging technique enables to photographically evaluate the presence and the location of posterior staphyloma. Thus, the study demonstrated that merely the presence of posterior staphyloma gave the possibility of eye disorders, whereas the location of posterior staphyloma was irrelevant to them. According to these results, when posterior staphyloma is not observed in high myopia patients with the ultra-widefield imaging, further careful investigation of the fundus with pupil dilation can be recommended to investigate peripheral degeneration and retinal breaks. Since patients with high myopia and no remarkable symptoms do not consult ophthalmologists so often, the expansion of application of this imaging technique could be of much help to provide possibilities of disorders related to pathological myopia. Considering these values of this study, we revised the description in the discussion section as follows.
Line 224:
Deleted; Our study suggested that ultra-widefield imaging could be applied to the evaluation and the prognosis of high myopia.
Revised; It is crucial to detect such abnormalities as early as possible in high myopia eyes because advances in myopic control have been remarkable. The results of the study indicate that when posterior staphyloma is not observed in high myopia patients with the ultra-widefield imaging, further careful investigation of the fundus with pupil dilation can be recommended to investigate peripheral degeneration and retinal breaks. Especially for the patients with high myopia, this imaging could be useful to provide possibilities of disorders related to pathological myopia.
Point #4
- The study looks like pure statistical analysis without much in-depth discussion about how the findings could be useful.
Response: We greatly appreciate your suggestion. As you pointed out, this study initially seems to have put importance on the statistical analysis. We reviewed and reconsidered the entire study and detailed the usefulness of the results, in which the ultra-widefield imaging technique could give us information of fundus disorders by analyzing the findings of posterior staphyloma. In addition, after visualizing the results by depicting the graphs from the tables as you suggested above, these results became much more understandable and made us aware other important findings. We added such findings in the discussion section along with the additional description made in the Point #3.
Line 215:
Added; This study demonstrated the superiority of this imaging technique to show the presence and location of posterior staphyloma. Furthermore, the relationship the findings posterior staphyloma and ocular disorders which are seen in pathological myopia raised the possibility of application of this imaging to clinical settings. It is crucial to detect such abnormalities as early as possible in high myopia eyes because advances in myopic control have been remarkable. The results of the study indicate that when posterior staphyloma is not observed in high myopia patients with the ultra-widefield imaging, further careful investigation of the fundus with pupil dilation can be recommended to investigate peripheral degeneration and retinal breaks. Especially for the patients with high myopia, this imaging could be useful to provide possibilities of disorders related to pathological myopia.

Round 2
Reviewer 1 Report
I have no further comments.